# Exploring the Impact of Charging Behavior on Transportation System in the Era of SAEVs: Balancing Current Request with Charging Station Availability

Yi Zhu [1], Xiaofei Ye [2,*], Xingchen Yan [3], Tao Wang [4], Jun Chen [5] and Pengjun Zheng [2]

1   Faculty of Maritime and Transportation, Ningbo University, Fenghua Road 818#, Ningbo 315211, China; zhuyi19991125@163.com
2   Ningbo Port Trade Cooperation and Development Collaborative Innovation Center, Faculty of Maritime and Transportation, Ningbo University, Fenghua Road 818#, Ningbo 315211, China; zhengpengjun@nbu.edu.cn
3   College of Automobile and Traffic Engineering, Nanjing Forestry University, Nanjing 210037, China; xingchenyan.acad@gmail.com
4   School of Architecture and Transportation, Guilin University of Electronic Technology, Jinji Road 1#, Guilin 541004, China; wangtao@guet.edu.cn
5   School of Transportation, Southeast University, Nanjing 211189, China; chenjun@seu.edu.cn
*   Correspondence: yexiaofei@nbu.edu.cn

**Abstract:** Shared autonomous electric vehicles (SAEVs) can offer safer, more efficient, and more environmentally friendly real-time mobility services with advanced autonomous driving technologies. In this study, a multi-agent-based simulation model considering SAEVs' vehicle range and charging behavior is proposed. Based on real-world datasets from the Luohu District in Shenzhen, China, various scenarios with different fleet sizes, charging rates, and vehicle ranges are established to evaluate the impact of these parameters on parking demand, charging demand, vehicle miles traveled (VMT), and response time in the era of SAEVs. The results show there would be much more charging demand than parking demand. Moreover, a larger fleet size and longer vehicle range would lead to more parking demand, more charging demand, and more VMT while increasing the charging rate can dramatically reduce the charging demand and VMT. Average response time can be reduced by increasing the fleet size or the charging rate, and a larger vehicle range leads to longer response time due to the longer time spent recharging. It is worth noting that the VMT generated from relocating from the previous request destination to the origin of the upcoming request accounts for nearly 90% of the total VMT, which should be addressed properly with appropriate scheduling. A charging policy considering current requests and the availability of charging stations was proposed and verified in terms of reducing the response time by 2.5% to 18.9%.

**Keywords:** shared autonomous electric vehicles; charging behavior; multi-agent-based simulation model; vehicle miles traveled; response time





## 1. Introduction

Shared autonomous vehicles (SAVs), as a new mode of transportation that combines the sharing economy and autonomous driving technology, have gained significant attention from both commercial and academic communities [1,2]. Many scholars believe that SAVs will revolutionize the future of urban transportation and provide much more convenient, safer, and economical mobility services [3,4]. Rapidly increasing research and development efforts are contributing to the expected presence of SAVs on the road, particularly in terms of parking, which they can free from the constraint of distance due to their self-driving ability [5]. A further way to enhance the advantages of SAVs is by replacing traditional fuel-powered vehicles with electric ones, so here comes SAEVs [6–9]. With the gradual decrease in the price of electric vehicles and the continuous development of charging infrastructure [1], the prospects for SAEVs are becoming increasingly promising. However,

the limitations in vehicle range and the time required for recharging SAEVs can influence the planning and operation of the transportation system [10], making it crucial to consider these two new constraints when trying to model the transportation system in the era of SAEVs.

Currently, most charging stations and parking lots are built separately, but for SAEVs, parking and charging can occur simultaneously [11]. For SAEVs that provide real-time mobility services, the reasons for parking are (1) low travel demand at the current period which does not need such a great number of SAEVs driving on the roads and (2) their current battery level is satisfied which needs no recharging at charging stations. Moreover, most existing parking lots have not considered installing charging facilities or only partially modifying some spaces to fulfill the demand for recharging, mainly due to safety and economic concerns. Charging electric-powered vehicles is a time-consuming process that inevitably occupies charging station spaces for extended periods of time. Compared to SAEVs that are parked in parking lots with no charging needs, if vehicles could be parked at charging stations, it would couple both demands and allow for charging while parking. Therefore, it is essential to study the impact of charging behavior on parking demand.

Although there are lots of studies on SAVs and SAEVs about their impact on the transportation system [12–16], to the authors' best knowledge, there are few studies that incorporate the constraints of vehicle range and charging behavior into a simulation model. Likewise, research focusing on the impact of SAEVs on parking demand can hardly be found which indicates the omission of this vital issue. Since the relationship between charging-related indicators (e.g., vehicle range and charging rate) and traditional transportation metrics (e.g., parking demand, VMT, and awaiting time) is of great significance to the sustainable development of SAEVs service, and a suitable charging policy would certainly be an assistance to the SAEVs service performance, this study aims to fill these research gaps by:

1. Developing a comprehensive multi-agent simulation model that considers both vehicle range and charging behavior of SAEVs to reveal the relationship between SAEVs' fleet size, charging rate, vehicle range, parking demand, charging demand, VMT, and average response time in the era of SAEVs based on the output of the experiment using real-world datasets in Shenzhen, China;

2. Dividing the total VMT into different parts based on the different origins and destinations and analyzing the specific VMT parts that counted most to provide suggestions for reducing these parts;

3. Proposing a charging policy that considers the balance between current requests and the availability of charging stations, and reveals its effectiveness on these mentioned metrics.

The remainder of this paper is structured as follows. In the next Section, previous studies related to these topics were comprehensively summarized. Then, we give a brief description of the dataset we used and the specifications of the proposed multi-agent-based simulation model. Next, the results of different experiment scenarios are discussed, and finally, conclusions from the discussion are presented and suggestions for future work are given.

## 2. Previous Studies

Many works have been completed focusing on SAVs' fleet size to analyze their impact on urban transportation systems, especially on parking demand [17–20]. An agent-based simulation model was proposed to estimate the potential impact of SAV in terms of urban parking demand. They pointed out that 90% of parking demand would be eliminated at a relatively low market penetration rate of 2%, but there would be an increase in VMT because of empty cruising [3]. After that, SAVs' parking demand in the City of Atlanta was further examined using a discrete-event, agent-based simulation model and the results suggested that parking demand can be reduced by over 90% for households who would give up private vehicles and use SAVs. Researchers also pointed out that the shift of urban

parking demand to adjacent areas may result in larger VMT, more congestion, and longer response time [4]. This model was further developed by focusing on the future trajectories of reduced parking demand [21]. It was pointed out that in the most optimal SAV adoption scenario, the parking demand will decrease by over 20% after 2030, especially in core urban areas. Taking parking preference date into consideration, a discrete event simulation model was proposed using the example of the University of the West of England, Frenchay campus as a case study to examine the impacts of SAVs. The results indicated that the parking demand decreased dramatically, leaving over 2500 m$^2$ of existing parking space unused [22]. Considering the fact that reducing parking demand may cost another price such as the increase in VMT and road congestion, reference [23] analyzed such side effects of SAVs from a transportation analysis zones (TAZs) perspective based on a dynamic traffic flow simulator named SOUND. The results show consistency with previous research that parking demand was reduced the most in residence-dominant zones in terms of quantity and office-dominant zones in terms of proportion, at the cost of a dramatic increase in empty VMT generated because of SAVs' relocation behavior which would therefore result in road congestion. Reference [24] explored the effects of SAVs based on a data-driven modeling approach using the dataset from Langfang, China. The simulation experiments contain two kinds of sharing schemes called "ride-sharing" and "car-sharing", respectively. The results indicated these two schemes would reduce parking demand by reducing car ownership, but VMT would increase regardless of sharing schemes while car-sharing alone increases much more VMT than ride-sharing. In terms of SAV service performance, reference [25] developed an agent-based simulation model for SAVs based in the city of Amsterdam, the Netherlands. Three different proactive relocation strategies named "Demand-Anticipation", "Supply-Anticipation", and "Demand-Supply-Balancing" were introduced and analyzed with regard to passenger waiting times and operational efficiency. The results show that "Demand-Anticipation" leads to the highest waiting time while the "Demand-Supply-Balancing" leads to the most favorable results in waiting time and operational efficiency. While these studies have simulated the impact of SAVs on urban parking demand, to further enlarge the benefit of SAVs by replacing traditional fuel with electricity, it is necessary to take the vehicle range and charging behavior of these service providers into consideration to reveal the relationship between charging related parameters (e.g., vehicle range and charging rate) and traditional metrics (e.g., parking demand, VMT and waiting time). Also, even though the mentioned research has confirmed that the SAVs would increase the total VMT due to empty travel, limited research has divided the total VMT into different parts to find out which part counts the most. This should not be overlooked since a clear understanding of this issue would be of great significance for operators to increase the overall performance of the SAV fleet by reducing empty travel.

Combining the upward trend of vehicle electrification and the promise of automation comes SAEVs [26,27]. A regional, discrete-time, agent-based model was proposed to explore the management of a fleet of SAEVs and the results indicated that SAEVs can serve nearly all requests with an average response time between 7 and 10 min [2]. A novel agent-based simulation framework was developed for electric vehicles by researchers considering the queuing issue in the fast charging stations [28]. By simulating an adaptive strategy based on implicit communication through booking systems in the charging station, the experiment results verified the effectiveness of the proposed approach in terms of route planning and reduction in total travel times within the whole system. Using MATSim, a multi-agent modeling platform, simulated performance characteristics of the SAEV fleet serving travel requests across the Austin, Texas 6-county region. It had been pointed out that reducing charging time and increasing fleet size can lower the response time but improving the vehicle range did not appear to do the same [6]. In the meantime, in order to reveal the impacts of SAEV in regard to socioeconomic heterogeneity, reference [29] modeled this new kind of mobility service with different pricing schemes with the help of MATSim. The results indicated that compared with reducing fares, reducing travel time for customers plays a much more important role in SAEV service usage. They also

pointed out that women tend to use SAEVs for shorter trips in regard to gender. As for the environmental effects of this new travel mode, reference [30] extends a multimodal transport model to simulate an increase in the market share of EVs to reveal their impacts from the environmental perspective. Their work based on MATSim pointed out that even the reduction in emissions could be limited if only short trips were served by EV. The impact can be higher if the government is able to target users with longer trips, at the cost of an optimized deployment charging stations for the sustainable development of such mobility service. After that, by taking economic metrics into consideration, the previous model was modified and pointed out that SAEV with longer vehicle range and charging station with fast charging rate can not only provide the best service for travelers but also the most profitable choice for companies providing mobility services using SAEV [7]. An agent-based simulation of SAEVs was performed across the Rouen Normandie metropolitan area in France to explore the impacts of different charging types and vehicle battery capacities on service efficiency. The results indicated that a faster charging rate results in higher performance, which means shorter response time. They also reveal the importance of choosing the right battery capacity to avoid the overlaps between demand and charging peak times [8]. By coupling charging and repositioning events and verifying the rightness of this kind of synergy using an agent-based model, the response time was 39% lower after coupling these two events [9]. Focusing on the environmental impact of SAEV and the First-Mile-Last-Mile system, research results showed that improving vehicle range can provide a better service [1]. The charging dispatching problem is also important for the mobility service provided by SAEVs, reference [31] tries to address this issue through their proposed traffic simulation framework based on a simulator Simulation Urban Mobility (SUMO) to improve the efficiency of charging station usage and save time for SAEVs users. Various Deep Reinforcement Learning (DPL) algorithms were performed to verify the robustness of their developed framework. Reference [12] evaluated the impact of charging infrastructure on SAEVs' service performance using an open-source agent-based simulation platform MATSim. The charging station within the simulation was generated to minimize the distance between the demand point and the charging station. The simulation results indicated that the combination of faster charging and such charging station planning would perform better than other scenarios but the battery swapping station would be better. Combining agent-based simulation and hybrid algorithm, reference [32] first allocated the charging demand based on the simulation output and then, used the proposed algorithm to site and size charging stations to meet the charging demand. However, even though most of the aforementioned research has revealed the impact of SAEV on response time, the parking demand, which is another important parameter to consider when it comes to urban transportation management, is still omitted. Additionally, when it comes to the charging policy, existing studies paid little attention to the relationship between the requests and the availability of charging stations, which should not be overlooked if the service providers and policymakers hope to increase the fleet performance to meet the high travel demand in peak time.

## 3. Materials and Methodology

### 3.1. Data Description

The simulation experiments designed in this study are based on several real datasets, including the parking lot dataset and charging station dataset provided by the Shenzhen government (https://opendata.sz.gov.cn, accessed on 25 August 2023) and a dataset containing trajectory information of the taxis running within the city for one day (https://people.cs.rutgers.edu/~dz220/data.html, accessed on 25 August 2023). Due to the nature of these former two datasets, the parking lots and charging stations in this research are assumed to be separated. Therefore, the situation of parking lots with charging functions is not considered here. The research area was limited to a circle located at the center of Luohu District, with a two miles diameter not only to reveal the effect of the introduction of SAEVs on the city's critical area but also because of the fact that the selected area would

generate much more travel requests and the research of this kind of area is necessary. The pre-processing process of these datasets and the processed data are elaborated as follows.

### 3.1.1. Travel Data

The following Table 1 shows an example of the unprocessed taxi trajectory dataset.

**Table 1.** Overview of the Original Taxi Trajectory Dataset.

| Taxi ID | Time | Longitude | Latitude | Occupancy Status | Speed |
|---|---|---|---|---|---|
| 34745 | 20:27:43 | 113.8068 | 22.62325 | 1 | 27 |
| 34745 | 20:24:07 | 113.8099 | 22.6274 | 0 | 0 |
| . . . | . . . | . . . | . . . | . . . | . . . |
| 28265 | 21:35:13 | 114.3215 | 22.7095 | 0 | 18 |
| 28265 | 9:08:02 | 114.3227 | 22.6817 | 0 | 0 |

Note: For Occupancy Statue, 1-with passengers and 0-without passengers. And the unit for Speed is mile/h.

The dataset includes a total of 1,155,654 pieces of records. According to Table 1, the dataset does not give the specific origin and destination of these trips, instead, the GPS location information and the corresponding instantaneous speed of the vehicle at certain points are given. In order to transform this dataset into a usable travel dataset, TransBigData, an open-sourced toolkit (https://transbigdata.readthedocs.io/en/latest/index.html, accessed on 25 August 2023) is used to exclude the anomalous data and to obtain the specific origin and destination of these trips. Fours steps including

1. Filtering to exclude data outside the research area;
2. Excluding data with instantaneous changes in Occupancy Status;
3. Rasterizing the GPS data and counting the amount of data in each raster;
4. Extracting the origin and destination points from the GPS data are taken before 13,380 pieces of OD (Original-Destination) data are shown in Table 2.

**Table 2.** Overview of the OD dataset after TransBigData.

| Trip ID | Start Time | Start LON. | Star LAT. | End Time | End LON. | End LAT. |
|---|---|---|---|---|---|---|
| 0 | 0:19:41 | 114.013016 | 22.664818 | 0:23:01 | 114.0214 | 22.663918 |
| 1 | 0:41:51 | 114.021767 | 22.6402 | 0:43:44 | 114.02607 | 22.640266 |
| . . . | . . . | . . . | . . . | . . . | . . . | . . . |
| 13378 | 23:03:45 | 114.118484 | 22.547867 | 23:20:09 | 114.133286 | 22.61775 |
| 13379 | 23:36:19 | 114.112968 | 22.549601 | 23:43:12 | 114.089485 | 22.538918 |

Note: Start and End refer to origin and destination, respectively, while LON. and LAT. refer to longitude and latitude, respectively.

### 3.1.2. Parking Lots Data

The original parking lot dataset contains 716 off-street parking lots in Luohu District, Shenzhen, with a total of 111,845 parking spaces. In the process of spatial visualization of these parking lots, it is found that not all the parking lots in the dataset are located within the Luohu District, therefore, the parking lots outside of Luohu District are excluded by using the tbd.clean_outofshape function in the TransBigData toolkit. After manually removing the parking lots with the help of a visualization map, a total of 308 parking lots with 3557 parking spaces within the study area were finally selected as shown in the following table (Table 3).

**Table 3.** Overview of the parking lot dataset after TransBigData and manual modification.

| ID | Price (RMB) * | Longitude | Latitude | Capacity |
|----|---------------|-----------|----------|----------|
| 1 | 5 | 114.1408886 | 22.5565399 | 8 |
| 2 | 5 | 114.1378183 | 22.5588927 | 6 |
| . . . | . . . | . . . | . . . | . . . |
| 307 | 5 | 114.112597 | 22.5791201 | 25 |
| 308 | 15 | 114.1068256 | 22.5834787 | 20 |

* Note: The exchange rate between RMB and USD on the day of data collection (2023.08.25) is 0.1385 which indicated that 1 RMB equal to 0.1385 USD.

The following figure (Figure 1) shows the spatial distribution of parking lots according to their parking capacity and parking price.

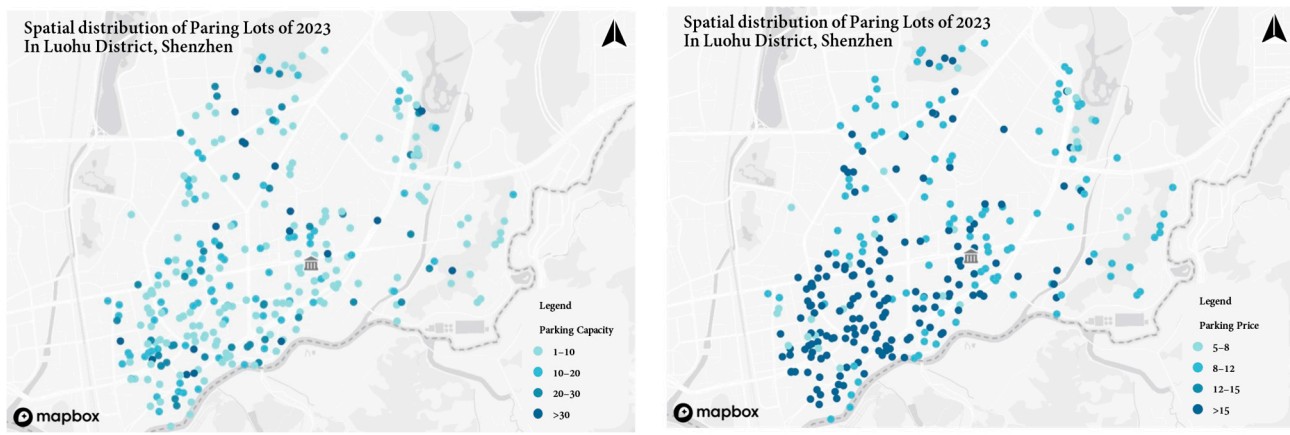

**Figure 1.** Parking capacity structure (**Left**) and parking price structure (**Right**) in Luohu District, Shenzhen.

### 3.1.3. Charging Stations Data

The initial charging station dataset includes 904 charging stations, providing 11,034 charging spaces. However, it only contains the address information of the stations, and there is no GPS location, which is not friendly to the subsequent experiments. Therefore, by using a Python script and the open-source Gaode API, these addresses are converted into points with GPS location information. After the tbd.clean_outofshape step, the number of charging stations within the study area was ultimately determined to be 267 with 2075 charging spaces, as shown in Table 4, and the spatial distribution of these stations according to their charging capacity is shown in Figure 2.

**Table 4.** Overview of the charging station dataset after TransBigData and manual modification.

| ID | Property | Longitude | Latitude | Capacity |
|----|----------|-----------|----------|----------|
| 1 | public | 114.117447 | 22.545063 | 2 |
| 2 | public | 114.118273 | 22.543836 | 2 |
| . . . | . . . | . . . | . . . | . . . |
| 266 | public | 114.181697 | 22.559031 | 10 |
| 267 | public | 114.183742 | 22.612863 | 4 |

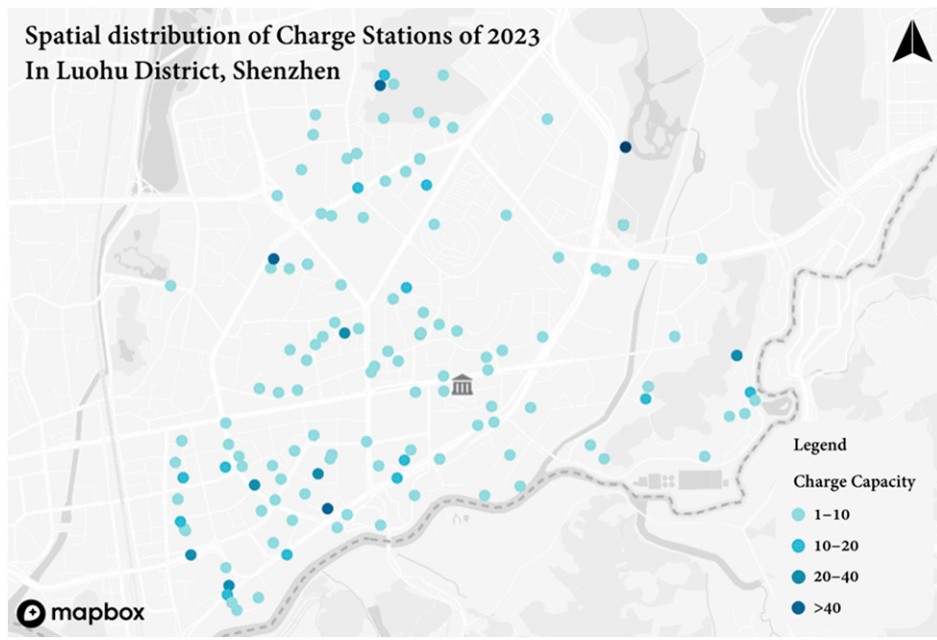

**Figure 2.** Charging capacity structure in Luohu District, Shenzhen.

### *3.2. Multi-Agent-Based Model Specification for SAEVs' Charging and Parking*

The proposed multi-agent simulation model, based on Anylogic, contained four kinds of agents: Traveler, SAEV, Parking Lot, and Charging station. The whole system can be summarized in Figure 3, the description of all states defined for these four kinds of agents can be found in Table 5, and the transitions between these states are shown in Figure 4.

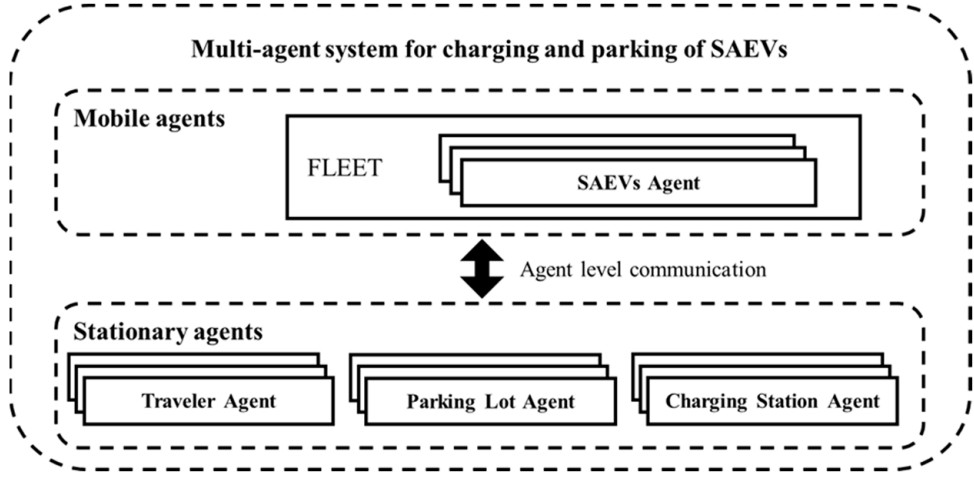

**Figure 3.** Overview of the proposed multi-agent simulation model.

**Table 5.** State description of the multi-agent simulation model.

| Agent | State | Description |
|---|---|---|
| Traveler | WAIT TO DEPART | Initial state of a Traveler agent waits for departure. |
| | WAIT FOR SERVICE | The Traveler's request has been successfully assigned to a SAEV and it is waiting for the arrival of this specific SAEV. |
| | WAIT FOR MATCH | The Traveler's request cannot be assigned to a SAEV and it has to be waited until there is an available one. |
| | TO DESTINATION BY SAEV | The Traveler is now taken to its destination by a SAEV. |
| SAEV | IN THE INITIAL SPOT | Initial state of an SAEV agent where it waits for requests. |
| | DRIVING TO PICK-UP | The state that the SAEV is driving to the pick-up point of the matched request. |
| | AT PICK-UP POINT | The state that the SAEV has arrived at the pick-up point of the matched request. |
| | DRIVING TO DROP-OFF | The state that the SAEV is driving to the drop-off point of the matched request. |
| | AT DROP-OFF POINT | The state that the SAEV has arrived at the drop-off point of the matched request and checking the battery status. |
| | AVAILABLE TO SERVE | The state indicates that battery level is sufficient for the upcoming request. |
| | DRIVING TO TARGET PARKING LOT | The state indicates that the SAEV is driving to the target parking lot. |
| | AT PARKING LOT | The state indicates that the SAEV has arrived at its target parking lot. |
| | DRIVING TO CHARGE STATION | The state indicates that the SAEV is driving to a charging station. |
| | AT CHARGE STATION | The state indicates that the SAEV has arrived at the charging station. |
| Parking Lot | PARKING AVAILABLE | The state indicates that there is an available parking space in the parking lot. |
| | FULLY OCCUPIED | The state indicates that there is no available parking space in the parking lot. |
| Charging Station | CHARGING AVAILABLE | The state indicates that there is an available charging space in the charging station. |
| | FULLY OCCUPIED | The state indicates that there is no available charging space in the charging station. |

### 3.2.1. Traveler Agent

For each *Traveler*, its initial state at the beginning of the simulation is called "WAIT TO DEPART". As the simulation proceeds, the *Traveler* generates a travel request based on the OD dataset, and the generated request will be instantly assigned to the nearest available SAEV. After the request has been successfully assigned, the Traveler changes its state to "WAIT FOR SERVICE" before the matched vehicle arrives. After the SAEV arrives and confirms the destination of this specific request, the *Traveler*, again, changes its state to "TO DESTINATION BY SAEV". If the request cannot be instantly assigned, in other words, all SAEVs are busy with requests that have been assigned to them before, the *Traveler* would update its state to "WAIT FOR MATCH" and the request will be stored into a collection called "Request list", which will automatically remove these requests once there is an available SAEV according to the First-In-First-Out principle. The time spent by the *Traveler* waiting for the arrival of a SAEV was called "Response time", which is an important indicator to evaluate the efficiency of the current transportation system. After the *Traveler* arrives at its destination, the SAEV completes its service and the travel request is satisfied.

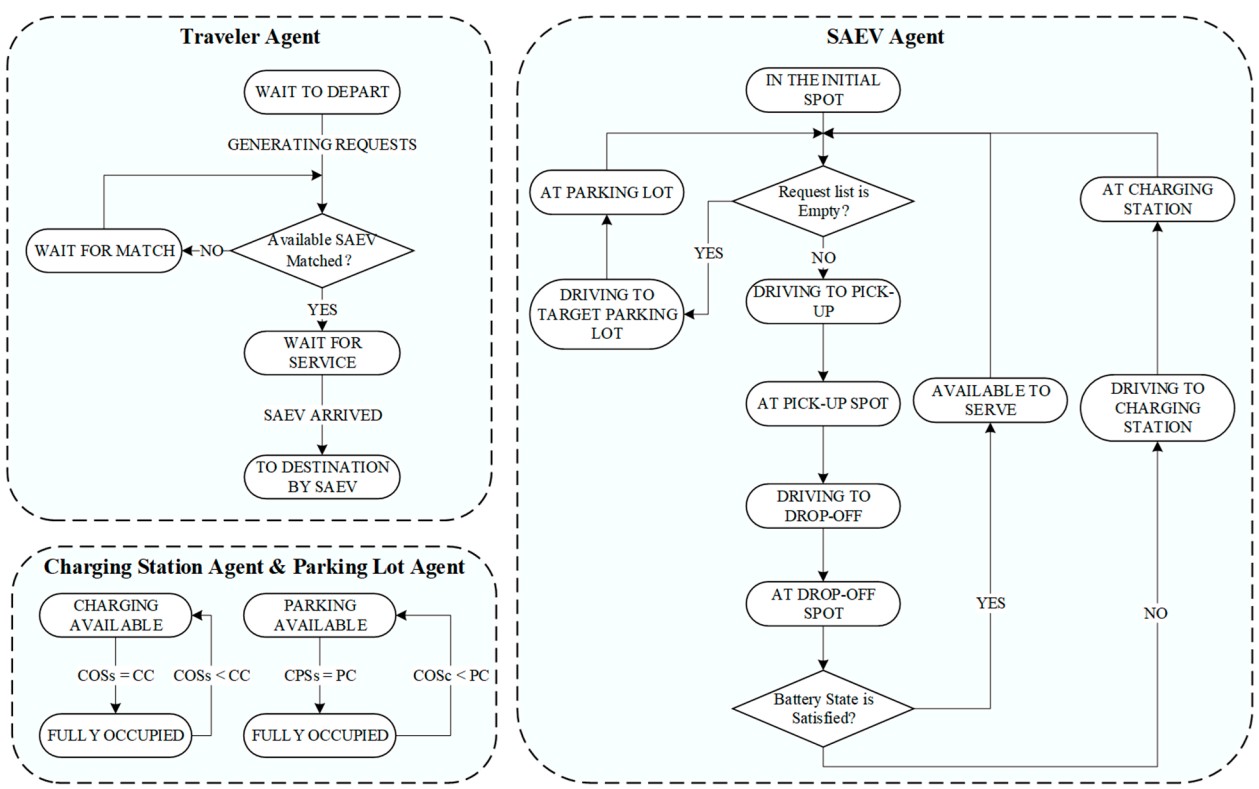

**Figure 4.** Individual agent actions in the multi-agent simulation model.

3.2.2. SAEV Agent

There are three phases for the *SAEV* agent, servicing phase, parking phase, and charging phase. At the beginning of the simulation, the *SAEV* are placed in the initial spot, a spot located at the center of the research area to indicate that all SAEVs start their daily work from here, and their state starts at "IN THE INITIAL SPOT".

The servicing phase begins after receiving the message "CHECK REQUEST QUEUE" from the system, the *SAEV* starts to check the collection called "Request list" which is used to store requests that need to be served. If the collection is empty, the parking phase begins, in order to reduce the cost of parking behavior and benefit the SAEV service provider economically, thanks to their self-driving ability, the *SAEV* would drive to $PL_{target}$ which can be chosen using the following Formulas (1) and (2):

$$PL = \begin{cases} PL_{avail} & if\ CPSs\ <\ PC \\ PL_{full} & if\ CPSs = PC \end{cases} \tag{1}$$

$$PL_{target} = \min_{cost}\{PL_{avail}\} \tag{2}$$

where,

$PL$ denotes the collection formed by all parking lots within the research area;

$PL_{avail}$, $PL_{full}$ indicate two collections that contain available parking lots and fully occupied ones, respectively;

$CPSs$ refers to current parked spaces, $PC$ refers to parking capacity;

$PL_{target}$ denote the specific parking lot the *SAEV* would drive to;

$C_{total}$ indicates the cost of choosing these parking lots as a target one at that specific simulation time, which can be calculated by Equation (3) where $C_{pp}$ indicates the price of the parking lot, $C_{ef}$ refers to the energy fee caused due to the travel to the parking lot [33] while $C_{rt}$ represents the road toll charged driving there. $l$ indicates the distance from the current location of the *SAEV* to the target parking lot, $\delta$ denotes the unit price of energy

consumed by the SAEVs, in this case, the value is pre-set as 10, $t$ represents the time spent to the parking lot and $\varepsilon$ was the unit road toll with a pre-set value of 7:

$$C_{total} = C_{pp} + \underbrace{14.58 \times l \times \delta \times (l/t)^{-0.68}}_{C_{ef}} + \underbrace{\varepsilon \times l}_{C_{rt}} \qquad (3)$$

Once the *SAEV* starts to drive to $PL_{target}$, its state would switch to "DRIVING TO TARGET PARKING LOT", at the same time, the value of CPSs in $PL_{target}$ would be increased by 1. At the time the *SAEV* arrives at $PL_{target}$, its state updates to "AT PARKING LOT", the system will count this specific moment as the start of a parking demand and record the time when the *SAEV* leaves the parking lot to accumulate the parking demand. But if the "Request list" is not empty, which means that there are requests that need to be served, then the servicing phase continues and the nearest available SAEVs (the *SAEV* that has not been matched to a request or the *SAEV* in the state called "AVAILABLE TO SERVE" to each of these requests will be assigned to them and update their state to "DRIVING TO PICK-UP". Once the *SAEV* arrives at the specific pick-up point, its state update to "AT PICK-UP POINT", this is the state that confirms the destination of this specific request. "DRIVING TO DROP-OFF" is the state after the confirmation of the trip destination and by the time it arrived at the drop-off point, its state changes to "AT DROP-OFF POINT". At the drop-off point, the *SAEV* will receive a message called "CHECK BATTERY STATE" from the system, and it will check its battery level, if the remaining power can cover the pre-set minimal range (in this study, 20%), its state will transfer to "AVAILABLE TO SERVE" to get ready for the next upcoming service and start checking the "Request list" again. But if the remaining power is lower than the minimal range, the charging phase begins. The *SAEV* will drive to $CS_{target}$ for charging and switch its state to "DRIVING TO CHARGING STATION". The $CS_{target}$ can be chose using Formulas (4) and (5):

$$CS = \begin{cases} CS_{avail} & if\ COSs\ <\ CC \\ CS_{full} & if\ COSs = CC \end{cases} \qquad (4)$$

$$CS_{target} = \begin{cases} \min_{occ}\{CS_{avail}\} & if\ Requetlist\ is\ Empty \\ \min_{dis}\{CS_{avail}\} & if\ Requestlist\ is\ not\ Empty \end{cases} \qquad (5)$$

where,

$CS$ denotes the collection formed by all charging stations within the research area;

$CS_{avail}$, $CS_{full}$ indicate two collections that contain available charging stations and fully occupied ones, respectively;

$COSs$ refers to current occupied spaces, $CC$ refers to charging capacity;

$CS_{target}$ denotes the specific charging station the *SAEV* would drive to; and,

$occ$ and $dis$ indicate the current occupied spaces of the charging station and the distance from the position of *SAEV* to the charging station, respectively.

The above charging policy considers the relationship between the current requests and the availability of charging stations. In order to reduce the response time of traveler, when there is a request waiting to be served, the *SAEV* would drive to the closest charging station to reduce the time spent on driving. However, if the request list is empty, the *SAEV* would choose the charging station with the least occupied spaces to increase the possibility of an available charger when it arrives and increase the utilization rate of existing charging facilities.

Accordingly, the $COSs$ of $CS_{target}$ would be increased by 1. When the *SAEV* arrives and starts charging, its state changes to "AT CHARGING STATION" and similar to before, the

system counts this specific moment as the start of a charging demand. The charging rate, *CR*, and the expected charged level, *ECL*, can be decided by Formulas (6) and (7), respectively:

$$CR = \begin{cases} GC & \text{if ``Request list'' is empty} \\ FC & \text{if ``Request list'' is not empty} \end{cases} \tag{6}$$

$$ECL = \begin{cases} TotalRange & \text{if ``Request list'' is empty} \\ AvailRange & \text{if ``Request list'' is not empty} \end{cases} \tag{7}$$

where,

*GC*, *FC* refer to the general charging rate and fast charging rate, respectively;

*TotalRange* indicates the *SAEV* would not leave the charging station until it is fully charged (100%) and, *AvailRange* indicates the *SAEV* would leave the charging station once it is charged to the pre-set value, which is 80% in this research.

As soon as the *SAEV* drives out of $CS_{target}$, its *COSs* would accordingly decrease by 1 and the system accumulates the value of this finished charging demand. The recharged *SAEV* will check the "Request list" after leaving $CS_{target}$ and drive to serve if there are any requests waiting, otherwise, they make a parking choice again. The final state of *SAEV* is "AT PARKING LOT", indicating all requests have been served and all SAEVs are within satisfied battery level.

### 3.2.3. Charging Station Agent and Parking Lot Agent

There are only two states in each *Charging station* agent, called "CHARGE AVAILABLE" and "FULLY OCCUPIED", respectively. The transactions between these two states are decided by one parameter called charge capacity (CC) and one variable called COSs. When the value of COSs is equal to CC, which means that all charging spaces of the charging station are occupied, and the state changes to "FULLY OCCUPIED", when there is a vehicle leaving the charging station, the state changes to "FULLY OCCUPIED". Similar to the *Parking Lot* agent, the two states are called "PARKING AVAILABLE" and "FULLY OCCUPIED", as shown in the figure above. When the value of CPSs is less than PC, the parking lot agent updates its status to "PARKING AVAILABLE". When the value of CPSs is equal to PC, the state changes to "FULLY OCCUPIED". Apart from these parameters and variables that affect state changes, there are two variables for both these two kinds of agents called "charging time" and "parking time", respectively, to reflect the charging demand and parking demand at each charging station and parking lot.

### 3.3. Simulated Scenarios and Experiment Setting

Scenarios with different SAEVs' fleet sizes which in this case is equal to the ratio of the number of SAEVs to the number of total requests, vehicle range which refers to the driving capacity of a SAEV with 100% battery level, and charging rate are designed to evaluate their impact on urban parking demand, charging demand, VMT, and response time. Since SAEVs can provide car-sharing services, unlike the private conventional vehicle, there should be one vehicle for one request to serve the travel demand immediately, even a small penetration rate of SAEVs can serve all requests within a reasonable response time [1,7]. After several warm-up simulations, a 5% penetration rate for 13,380 trips (which has only 699 SAEVs) was chosen as the base case as the average response time can be maintained to a relatively short time between 5 to 8 min, and the value will constantly increase to 10% (1338 SAEVs) in 1% increments. Additionally, as the research area is not quite large, which only contains the traffic entities within 2 miles of the district center, we minimized the vehicle range accordingly. It was assumed that the vehicle range increased from 120 miles to 200 miles, in 20 mile increments. In order to simplify the calculation of SAEVs' charging time, mile/h is used as the unit of charging rate, which indicates the distance a SAEV can drive after pre-unit of time charging. The general charging rate varies from 30 mile/h through 50 mile/h, in 5 mile/h increments, while the fast charging rate varies

from 60 mile/h through 100 mile/h, in 10 mile/h increments [6]. Table 6 provides a summary of all pre-set parameters. Additionally, in order to verify the effectiveness of the proposed charging policy in reducing the response time and decreasing the total VMT while increasing the utilization of the existing parking and charging facilities, comparison experiments were conducted where the SAEVs would choose the target charging station simply based on the distance.

**Table 6.** Summary of all simulated scenarios parameters.

| Vehicle Range (mile) | 120 | 140 | **160** | 180 | 200 | NA |
|---|---|---|---|---|---|---|
| Charging rate (GC/FC) (mile/h) | 30/60 | 35/70 | **40/80** | 45/90 | 50/100 | NA |
| Fleet size (%) | **5** | 6 | 7 | 8 | 9 | 10 |

Note: GC refers to general-charging, FC refers to fast-charging. Base case parameters are shown in bold.

Apart from the above parameters set, we assumed that:

- All SAEVs were fully charged at the beginning of the simulation;
- If the remaining power of a SAEV is not enough to cover the distance to the nearest charging station, it will transfer to the target charging station within 0.01 s as soon as it runs out of power;
- Due to the size of the research area and the fact that even in a scenario with the largest fleet size, there would be only 1338 SAEVs within the network, which is a relatively small number of vehicles, so the effect of road congestion on SAEVs' speed is not considered throughout the whole process, in other words, the SAEVs maintain a fixed speed throughout the whole simulation.

The output indicators of the simulation experiment include the parking demand, charging demand, total VMT, and average response time. The former two are counted in hours and would be accumulated through the whole simulation period. The spatial distribution of these two kinds of demand will be demonstrated on the map. At the same time, the total VMT will be divided into different parts to find out which part accounts for the most by counting the exact time SAEV arrived at different origins and destinations. The total VMT can be calculated by Formula (8):

$$VMT_{TOTAL} = VMT\_I + VMT\_E + VMT\_P + VMT\_CS \tag{8}$$

where,

$VMT_{TOTAL}$ indicates the total VMT generated by a SAEV during the whole simulation period;

$VMT\_I$, $VMT\_E$, $VMT\_P$, and $VMT\_CS$ indicate the VMT generated by a SAEV starting from the initial point, start point, end point, parking lot, and charging station respectively. They can be calculated by Formulas (9)–(12):

$$VMT\_I = VMT\_IS + VMT\_IP \tag{9}$$

$$VMT\_E = VMT\_ENS + VMT\_EP + VMT\_ECS \tag{10}$$

$$VMT\_P = VMT\_PS \tag{11}$$

$$VMT\_CS = VMT\_CSS + VMT\_CSP \tag{12}$$

where,

$VMT\_IS$ and $VMT\_IP$ denote the VMT generated by a SAEV starting from the initial point to one start point or to one parking lot, the latter one can only take place at the very beginning of the simulation as the number of requests then is less than the number of idle SAEVs;

*VMT_ENS*, *VMT_EP* and *VMT_ECS* indicate the VMT generated by a SAEV starting from one endpoint to the next start point, the parking lot and the charging station, respectively;

*VMT_PS* indicates the VMT generated by a SAEV starting from one parking lot to one start point;

*VMT_CSS* and *VMT_CSP* indicates the VMT generated by a SAEV starting from one charging station to one start point and to one parking lot, respectively.

The average response time can be calculated by Formula (13):

$$T = \frac{\sum\limits_{n=1}^{m} t_2^n - t_1^n}{m} \tag{13}$$

where,

$t_1^n$ indicates the time the $n^{th}$ request was generated, $t_2^n$ indicates the time the matched SAEV (for request *n*) arrived, and *m* indicates the number of the total travel demand within the whole system, in this case, the value of *m* is 13,380.

The simulation will stop when all requests have been served and all the SAEVs have been parked at the parking lots.

## 4. Results and Discussion

### 4.1. Parking Demand and Charging Demand

The results of parking demand and charging demand in each scenario are shown in Figures 5 and 6 separately.

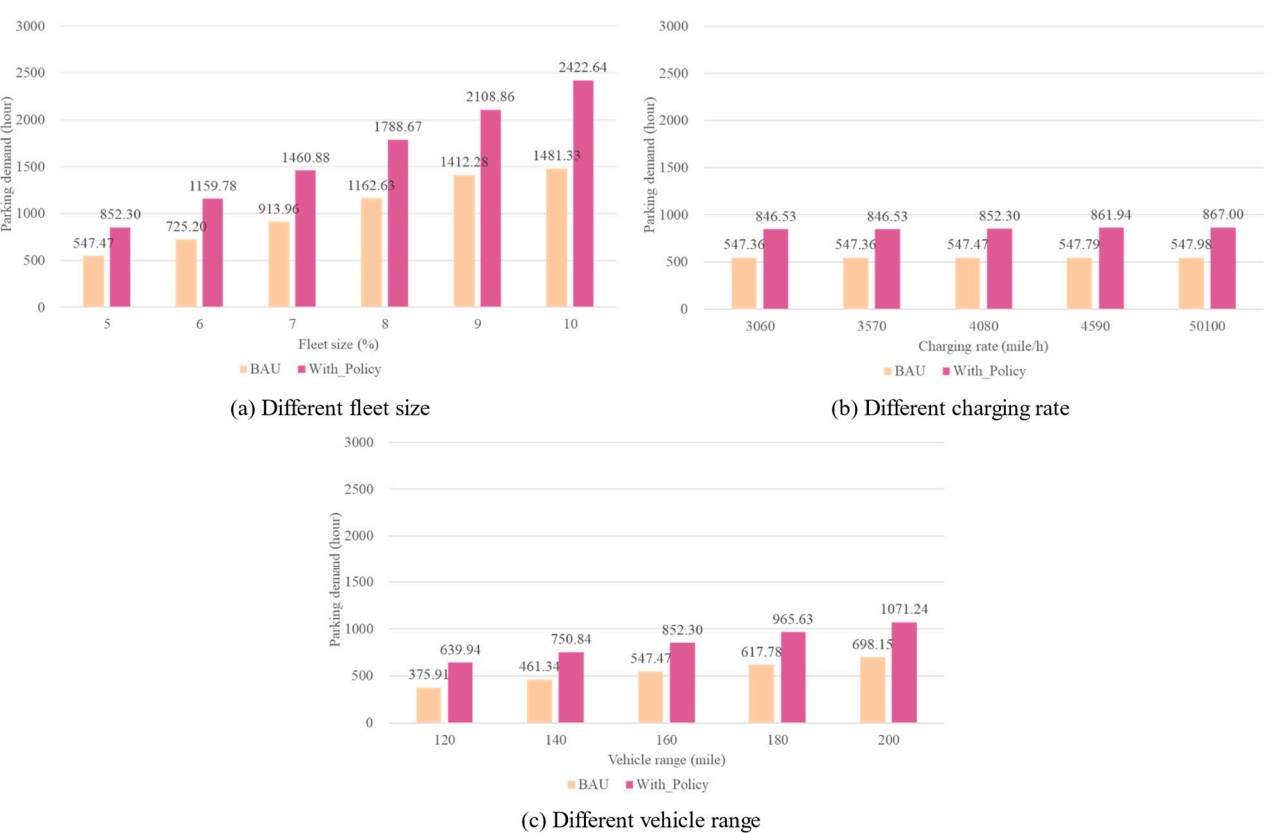

**Figure 5.** Overview of parking demand in various simulation scenarios.

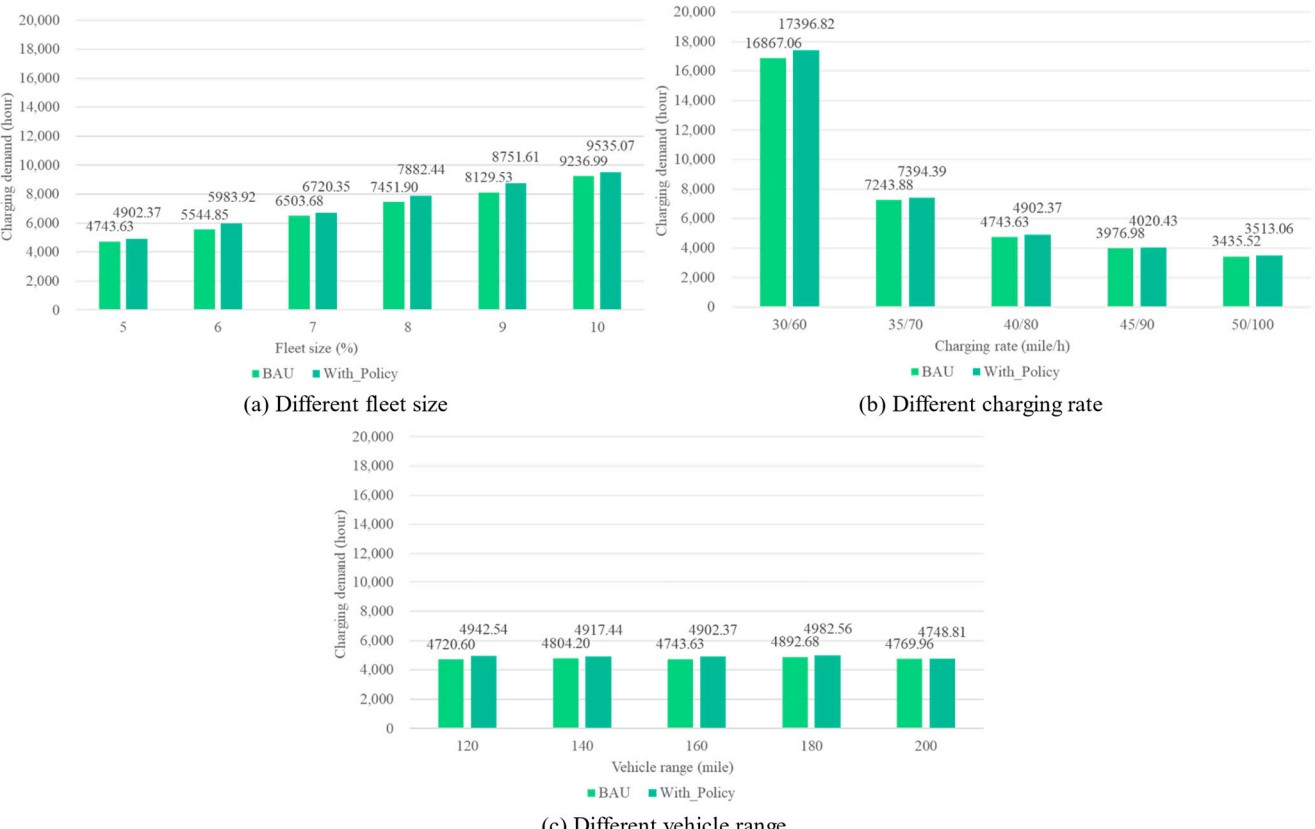

**Figure 6.** Overview of charging demand in various simulation scenarios.

According to Figures 5 and 6, as the fleet size continues to rise, the parking demand increases by nearly two times, from 547.47 h when the fleet size is only 5% (which is 669 SAEVs) to 1481.33 h with the maximized fleet size. The charging demand also saw an increase when the fleet size kept rising and the value doubled (from 4743 h to 9236 h). This is not unexpected as a larger fleet size means more SAEVs within the system are ready to serve and the travel request can be satisfied quicker, which increases the possibility of SAEVs becoming idle and choosing to park, and more SAEVs would inevitably increase the charging demand as the total number of SAEVs within the system that may need recharging has been increased, even though they may not drive to serve after charging. It is obvious that the charging demand in all scenarios (varying from 3436 h to 17,397 h) is much higher than the parking demand (376~2423 h). This is an expected result because compared with parking lots, charging stations can not only offer a place to park but also a place to recharge vehicles' batteries. Charging is a time-consuming process, which may also help these idle SAEVs that cannot find any requests at off-peak hours without doing nothing, in this case, recharging their batteries if necessary. For the car-sharing SAEVs, they can reduce their parking time by constantly serving different requests if there are any, for those that need recharging, the duration at the charging station can also be seen as the time waiting for the new requests, so it can be said that charging demand should be the one need to be concerned more compared with its parking counterpart after taking charging behavior and vehicle range into consideration.

Unlike the parking demand, which seems not to be effectively affected by the change in the charging rate, it is expected to see that the charging demand experienced a decrease as the charging rate increased (from 16,867 h to 3436 h). This is reasonable since a faster charging rate means less time spent in a charging station, which will shorten the time for SAEVs spent on charging and therefore increase the charging station's turnover efficiency and be seen as an effective way to reduce the time for SAEV to find an available charger. However, it is not possible to unlimitedly increase the charging rate since the reduction

become much smaller and may not be an economical way to do so, even though this study did not place any limitations on this value, there should be some due to safety and power resource allocation considerations.

Increasing the vehicle range causes an increase in parking demand, as the vehicle range increases from 120 miles to 200 miles, the parking demand increases from 376 h to 698 h, while the charging demand keeps the value constant at 4800 h. The increase in parking demand may be because a larger vehicle range means a wider service range, a SAEV can serve more travel needs without being recharged, which makes some other SAEVs idle for some time, and consequently need to choose a parking lot. The constant charging demand may be because the total miles traveled to serve these requests are firm, so the change in vehicle range may only affect which SAEV is chosen to serve the traveler, which would not pose a threat to the total generated charging demand.

The proposed charging policy would increase the parking demand and charging demand in all scenarios, while the increase for charging is less noticeable (maximal value is 8%) than parking (from 49% to 70%). This may come from taking charging availability into consideration when choosing a charging station, all SAEVs within the system have a higher level of power, the possibility of charging a SAEV is not so common compared with the need to park as there may not be queued request and none of these idle SAEV were need to be charged and would be ready to serve request whenever there are any.

*4.2. VMT*

The VMT generated by empty driving due to SAEVs' self-driving ability has been considered a great waste of road resources and energy (electricity for SAEVs). So, it is necessary to clarify the origin of VMT and control it accordingly.

As shown in Figure 7, the total VMT rises as the fleet of SAEVs increases, but the increment is not significant which indicates that although more SAEVs on the road network will lead to more VMT, the impact is not significant. Also, as the vehicle range increases, the total VMT within the system increases significantly. The reason for this increase is the increase in the *VMT_ENS* part, possibly because increasing the vehicle range will inevitably expand the SAEVs' service range and increase the possibility of long-distance migration within its service range for request servicing. On the contrary, as the charging rate increases, the total VMT decreases significantly, mainly due to a significant decrease in the *VMT_ENS* part. This noticeable reduction may be because a fast charging rate brings less charging time, and the former requests served by relocated SAEVs which are far away from the start points can now be assigned to closer SAEVs which missed these requests due to the longer charging time they previously spent in the charging stations. This kind of situation occurs quite frequently based on the dramatic decrease in *VMT_ENS*.

The major part of the total VMT generated is *VMT_ENS*, while the others only accounted for small portions. The increase in vehicle range leads to higher values in *VMT_PS*, mainly because a wider service range brought by higher vehicle range enables SAEVs to serve more requests without recharging, while these not close enough to requests must drive to parking lots. A possible reason why *VMT_CSP* increases, as the fleet size becomes larger, may be the increase in the number of vehicles capable of providing services within the system reduces the possibility of demand queues, making it more likely for fully recharged SAEVs to drive to park once they leave the charging station. The large amount of *VMT_ENS* (nearly 90%) in all BAU (Business as usual) scenarios indicates that although relocating SAEVs to the next service immediately after they completed the previous one can increase the vehicle utilization rate, it can also increase the VMT within the entire road network. Since the effect of road congestion is not considered in this research, in reality, the more vehicles on the road, the more likely it is to cause congestion. Therefore, it is necessary to have a reasonable scheduling and allocation scheme of vehicles to reduce long-distance migration, thereby reducing excessive empty VMT, energy consumption, and road occupancy.

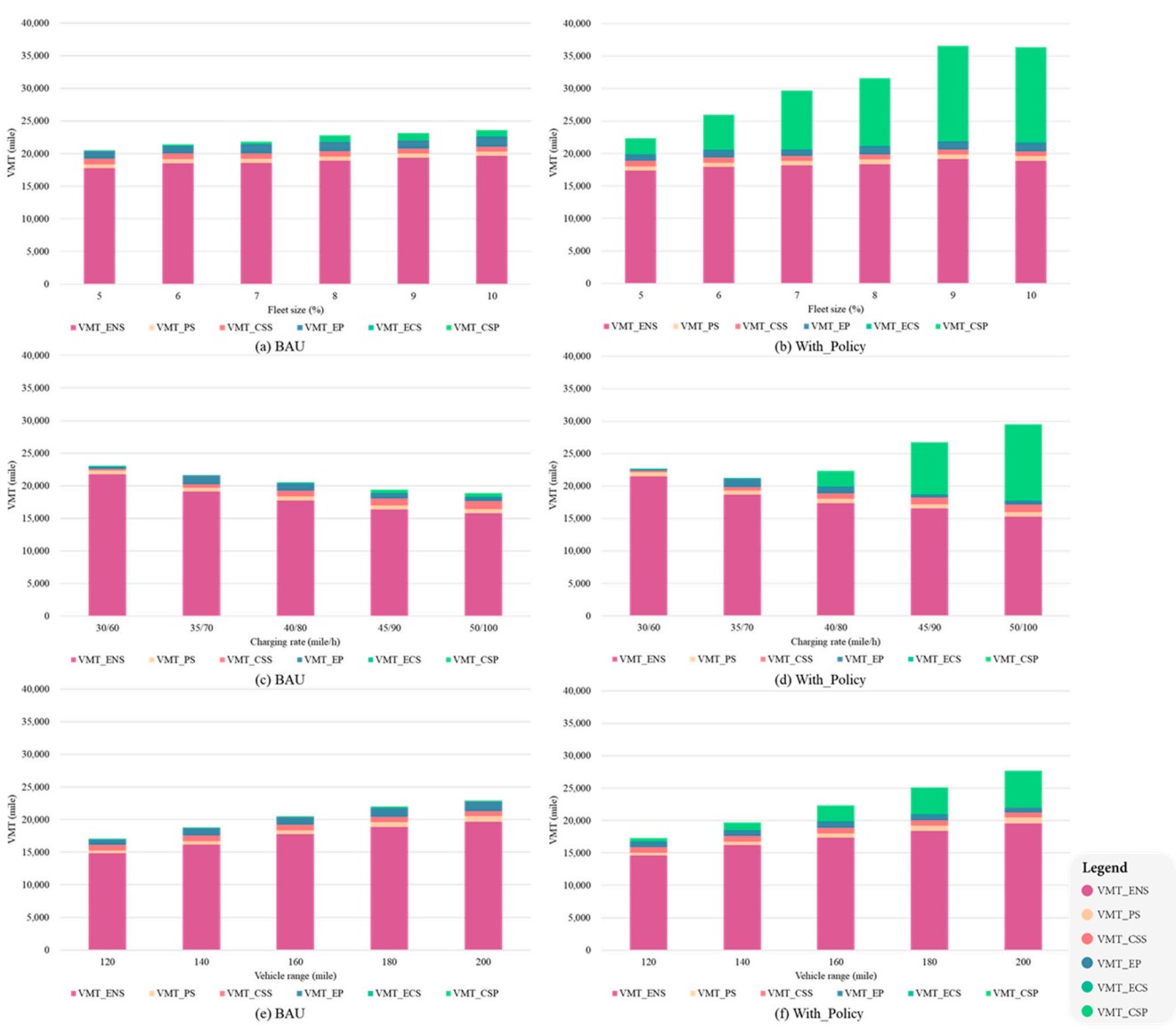

**Figure 7.** Overview of VMT in various simulation scenarios.

For scenarios with a charging policy, the total VMT is likely to show a rising trend as the fleet size and the vehicle range go up, similarly, even with some fluctuations, the total VMT also increase as the charging rate rises. The value of total VMT is much higher compared with BAU scenarios, mainly coming from the increase in the portion *VMT_CSP* part. This part of VMT also shows a positive relationship with fleet size, charging rate, and vehicle range. This is not unexpected because more SAEVs within the system means that there would be more vehicles ready to serve so during the charging process of some SAEVs, requests might be satisfied by other vehicles and these recharged vehicles may drive to park after finishing charging. A faster charging rate can reduce the time spent in the charging station and therefore enable the recharged SAEVs to return to the system quicker to serve the waited travel requests. Similarly, a larger vehicle range can ensure a longer time before the SAEV requires recharging, even if there were some of them below the expected battery level, others with the same vehicle range but not have been assigned to any requests can be the alternative when travelers need to be served, so after these recharged SAEVs out of the charging station, the only thing they have to do is drive to a parking lot.

### 4.3. Average Response Time

The average response time in different simulation scenarios can be found in Figure 8. As the fleet size increases, the average response time decreases from 7.99 min (5%) to 3.92 min (10%). This is reasonable as more SAEVs in the system lead to fewer orders being in the queue, therefore reducing the response time. However, it is worth noting that the reduction is becoming smaller which indicates that continuously increasing the number of vehicles in the transportation system can be an effective way to control the response time but the positive effect may be a threshold and an optimal value of the fleet size.

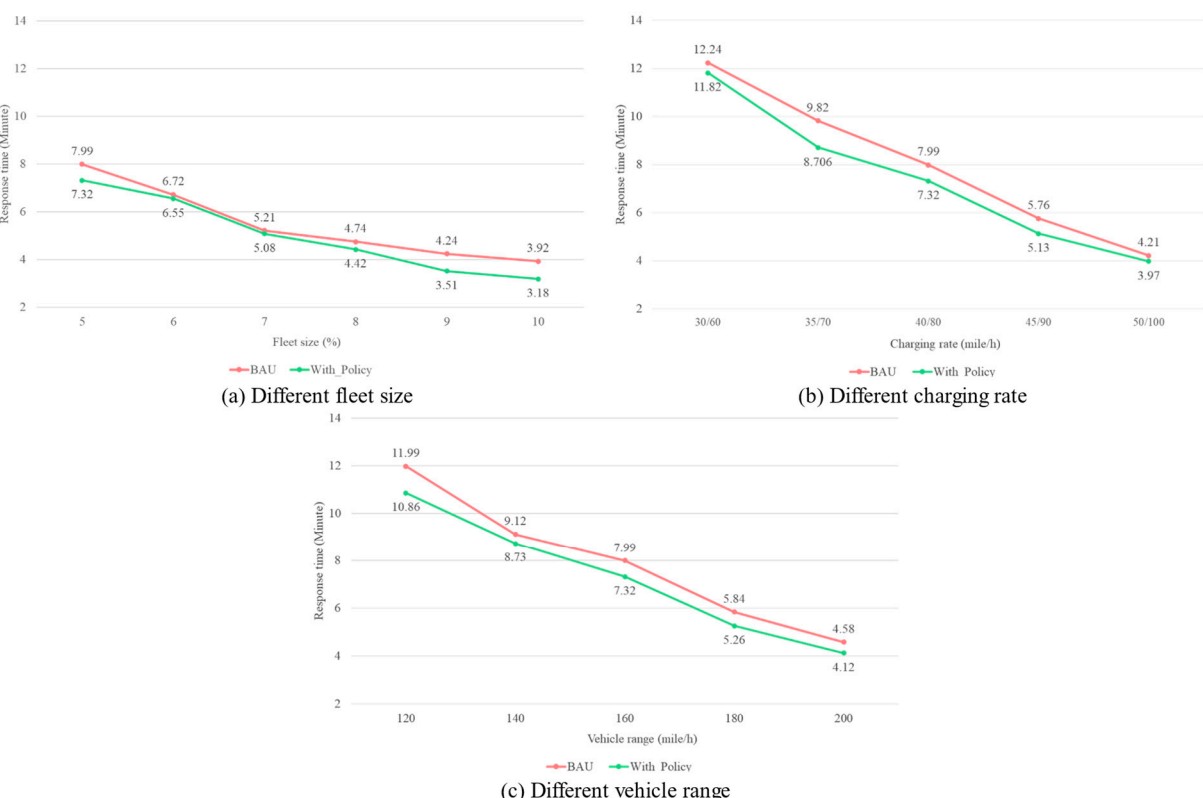

**Figure 8.** Overview of response time in various simulation scenarios.

As can be seen from the figure, the charging rate has a significant impact on the response time. The average response time at a charging rate of 30/60 miles/h (12.24 min) is almost 2 times longer than a charging rate of 50/100 miles/h (1.34 min). A faster charging rate can reduce the time SAEVs spend at charging stations and enable faster replenishment of available SAEVs into the service system. The change of vehicle range also shows a similar trend to the charging rate, the response time saw its peak value (which is 11.99 min) in a scenario with the smallest vehicle range, in this case, 120 miles, and as the vehicle range continue to rise, the response time reduces to 4.58 min. This can be explained from the point of view that before vehicles consume their entire battery capacity, vehicles with longer ranges can provide service with a shorter response time. After that, even a longer charging time would be required for SAEVs with longer vehicle range, the overall result of the simulation indicates that this would not affect the total response time within the whole simulation process, it is reasonable to say that vehicles with larger vehicle range can provide mobility service with a shorter response time. As it is not feasible to expect all SAEVs to complete all requests without recharging, therefore, to reduce response time, it is recommended to consider a combination of increasing the number of service vehicles, increasing the charging rate, and increasing the vehicle range.

It is also clear to see that in scenarios with the application of the proposed charging policy, the response time has been reduced to varying levels, from 2.5% to 18.9%. As the

charging policy considers the current requests and the availability of the charging station, on one hand, during the peak hour, the SAEVs would choose the closest charging station to prioritize the charging demand so that they can finish the charging process as quickly as possible in order to serve the request if there are requests waiting to be satisfied. On the other hand, SAEVs would like to be easily charged as they would choose a charging station with minimal occupied spaces when in off-peak hours, this would reduce the time spent on driving to an available charging station and increase the number of ready-to-serve SAEVs in the traffic system. Both of these two kinds of charging behaviors were designed to fulfill the charging demand as soon as possible, the only different is the former one does this when it has to, and the latter one does this in advance to avoid poor battery state. Therefore, it is reasonable to see a reduction in response time after the implementation of the policy.

## 5. Conclusions and Future Work

SAEVs can provide car-sharing mobility services, reducing the total number of vehicles within the entire transportation system which will directly reduce the need for parking and reduce the pollution by using environmentally friendly electric power. But the vehicle range and their charging behavior can sometimes become barratries to further benefit the transportation system. This research proposed a multi-agent-based simulation model that considers both vehicle range and charging behavior of SAEVs and reflects their real-time battery level by monitoring their driving mileage. Based on a real dataset from the Luohu District in Shenzhen, various scenarios with different fleet sizes, charging rates, and vehicle ranges are explored to evaluate the impact of SAEVs on parking demand, charging demand, total VMT, and average response time and relationship between these indicators are revealed, and a charging policy consider the current requests and the availability of charging station are proposed and verified. The results indicate that after considering the charging behavior of SAEVs, the charging demand within the whole system is much more than the parking demand as SAEVs only park when their battery level is satisfied and there are no requests waiting to be served. Larger fleet size and longer vehicle range can lead to more parking demand while changing the charging rate would not influence the parking demand. A larger fleet size would result in more charging demand and a faster charging rate, which can dramatically reduce charging demand, while the change in vehicle range would not have an impact on the charging demand. The total VMT will increase as the fleet size becomes larger or the vehicle range increases and will decrease as the charging rate increases. Moreover, a large portion of VMT is generated by SAEVs relocating from the destination of the last service to the origin of the next request. This portion of VMT can be reduced by rational scheduling of SAEVs within the network. The average response time would decrease as the fleet size goes bigger, the charging rate goes faster and the vehicle range goes larger. But the reduction caused by fleet size changes is not as remarkable as the other two, which indicates that unlimitedly increasing the number of SAEVs in the system will not always shorten the response time as we expected. Therefore, to improve the quality of mobility service in terms of response time, it is reasonable to combine the change in fleet size, charging rate, and vehicle range. To verify the effectiveness of the proposed charging policy, the results of the comparison experiments show that when considering the current requests and the availability of charging stations, the response time would be further reduced by 2.5% to 18.9% in different scenarios, the parking demand would increase as there would be more SAEVs to be idle and choose to park, the charging demand would not be influenced much; however, the total VMT would increase because a great number of VMT is generated during the SAEVs driving to the parking lot after recharge.

The main contribution of this paper is threefold. Firstly, a multi-agent-based simulation model considering the vehicle range and charging behavior was proposed to reveal the relationship between fleet size, charging rate, vehicle range, parking demand, charging demand, VMT, and average response time.Secondly, the VMT has been divided into parts according to the different origins and destinations of SAEVs during the whole simulation process to clearly the portion of each part. Thirdly, the proposed charging policy considers

the current request and the availability of charging stations was verified to be effective in reducing the response time.

However, there are some limitations of this research that should be addressed in future studies. First, the calculation of battery consumption and the recharging process should be more precise. Second, due to the poor performance of the used computer, the simulation experiment has narrowed down the research area to a specific scope, within a 2 mile radius of the Luohu District center, and the vehicle range has also been reduced accordingly. In future research, expanding the research area and using a more accurate vehicle range should be taken into consideration. Additionally, apart from fixed charging service providers such as charging stations mentioned here, a novel mobile charging approach provided by mobile charging vehicles should as be considered in future studies since the combination of these two can combine the advantages of both to serve SAEVs in dynamic and static ways [34,35], which may be the solution to a much more sustainable mobility system. Finally, this paper did not consider the impact of road congestion on vehicle speed, which should be incorporated into the transportation simulation model in future research.

**Author Contributions:** Conceptualization, Y.Z. and X.Y. (Xiaofei Ye); methodology, Y.Z.; software, Y.Z.; validation, Y.Z. and X.Y. (Xiaofei Ye); formal analysis, Y.Z.; resources, Y.Z.; data curation, Y.Z.; writing—original draft preparation, Y.Z.; writing—review and editing, Y.Z. and X.Y. (Xiaofei Ye); visualization, Y.Z.; supervision, X.Y. (Xiaofei Ye), X.Y. (Xingchen Yan), T.W., J.C. and P.Z.; funding acquisition, X.Y. (Xiaofei Ye), T.W. and P.Z. All authors have read and agreed to the published version of the manuscript.

**Funding:** This research was funded by the Fundamental Research Funds for the Provincial Universities of Zhejiang (SJLY2023009), Transportation Technology Plan Project of Ningbo, Zhejiang (202214), the National "111" Centre on Safety and Intelligent Operation of Sea Bridge (D21013), National Natural Science Foundation of China (Nos. 71971059, 52262047, 52302388, 52272334 and 61963011), the Natural Science Foundation of Jiangsu Province, China (no. BK20230853), the Specific Research Project of Guangxi for Research Bases and Talents [grant number AD20159035], in part by Guilin Key R&D Program [grant number 20210214-1], and Liuzhou Key R&D Program [grant number 2022AAA0103].

**Data Availability Statement:** Data used in this research can be found through these links provided in Section 3.1 Data description.

**Acknowledgments:** The authors thank their mentors who provided instructions on writing this paper.

**Conflicts of Interest:** The authors declare no conflict of interest.

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
