# Peer review of "Exploring the Impact of Charging Behavior on Transportation System in the Era of SAEVs: Balancing Current Request with Charging Station Availability"

_systems, doi:10.3390/systems12020061_

Round 1
Reviewer 1 Report
Comments and Suggestions for Authors
1. The current literature review section cites too few references and does not discuss the latest research, such as "On the simulation of shared autonomous micro-mobility." The authors should significantly supplement with recent studies to highlight the contributions of this paper.
2. The summary of contributions should be as concise as possible to allow readers to intuitively understand the novelty of the paper.
3. The font size in the figures should be uniform. The fonts in Figures 2 and 3 are too large, while those in Figures 4 to 8 are too small.
4. The formatting of formulas should be adjusted according to the standards; for example, variables in formulas (such as $n$, $m$ in Eq. (13)) should be in italics, and the multiplication symbol should be $\times$ instead of $*$.
5. "RMB" is not the international standard notation for currency.
6. Would the introduction of mobile charging proposed in “The parallel mobile charging service for free-floating shared electric vehicle clusters” and “The multi-mode mobile charging service based on electric vehicle spatiotemporal distribution” change charging behavior? The authors should discuss this.
Comments on the Quality of English LanguageEnglish is okay but can be improved further
Author Response
Please kindly see the attached file. Your cooperation and understanding would be highly appreciated.

Reviewer 2 Report
Comments and Suggestions for Authors
Exploring the Impact of Charging Behavior Consider the Balance between Current Request and Availability of Charge Station on the Transportation System in the Era of SAEVs
Overall evaluation
==================
The paper considers several scenarios with respect to Shared Autonomous vehicles characteristics and evaluate the need for charging stations, parking lots and quality of services for the users (response time).
A multiagent simulation is conducted with a good data source is used to model
the origin destination matrix.
The topic is interesting and the paper is not difficult to read. Most ideas are
clear but several shortcoming are adopted with make the reader aware of their limitation. I hope the reader focus on this point for the revised version.
Main issues
===========
The simulations are based on discrete events (DE) methodology. This is not standard
transportation science, where specific transport simulators are used.
The problem with DE tools is that congestion, a fundamental feature of traffic,
is not considered. Congestion is important in this context, because vehicle travelling to
charging stations create an induced traffic producing delays to other vehicles.
If these vehicles run on fossil fuels, this can be a source of pollution that can limit the
benefits of electric vehicles. The author may compare with analyses conducted on
multi-agent simulations and deployment of charging stations, for example:
- De Wolf, D., Diop, N., & Kilani, M. (2022). Environmental
impacts of enlarging the market share of electric
vehicles. Environmental Economics and Policy Studies.
- García-Magariño, I., Palacios-Navarro, G., Lacuesta, R., &
Lloret, J. (2018). ABSCEV: An agent-based simulation
framework about smart transportation for reducing waiting
times in charging electric vehicles. Computer Networks
- Müller, J., Straub, M., Naqvi, A., Richter, G., Peer, S.,
& Rudloff, C. (2021). MATSim Model Vienna: Analyzing the
Socioeconomic Impacts for Different Fleet Sizes and
Pricing Schemes of Shared Autonomous Electric Vehicles.
- Song, Y., Zhao, H., Luo, R., Huang, L., Zhang, Y., & Su,
R. (2022). A sumo framework for deep reinforcement
learning experiments solving electric vehicle charging
dispatching problem. arXiv preprint arXiv:2209.02921.
The literature review focuses on shared autonomous vehicles but the authors do not consider
the rich literature on traffic simulation. This should be addressed (the authors may start
with the references above).
Specific issues
===============
- The title is too long
- Lines 23-25: the results are as expected; could be mentioned in the text
- "response time" used 14 times in pages 1, 2 and 3 is only defined in page
eight (Line 213)
- Use "charging station" instead of "charge station"
- Equation (2) is a strong assumption. It is more empirical sound to use a discrete choice model (such us the logit model for the choice behavior). Please comment your assumptions.
- Line 233, please use a good math notation so that "cost" appear as in equation (3)
- Where do the numerical values in Equation (3) come from ?
- In Equation (13) make it clear that t_1 and t_2 depend on n. Otherwise the summation is not necessary
- Line 266-267: check punctuation and add ", respectively." at the end
- Table 6: check labels (First row : Traveled distance? Third row: Penetration rate of SAEV ?)
Comments on the Quality of English LanguageOverall I find it OK, minor check should be enough.
Author Response

(The authors gave the same response as above.)

Round 2
Reviewer 1 Report
Comments and Suggestions for Authors
Thanks to the authors for responding to my review comments.
Author Response
Thanks for your help during the revision of this manuscript. Your time and generosity are highly appreciated.
Reviewer 2 Report
Comments and Suggestions for Authors
The revised version is better than the initial one, but it seems that the authors have submitted the manuscript before read it careffully. Further cleaning is required.
Examples:
- Response to my question number 9 is OK, but the text in the manuscript is not revised as indicated by the authors.
- line 433: "and" is used bewteen t_1 and t_2, but a period "," is used for "m" which comes next. Please check consistency.
- the references I have suggested were added as is, but notice that a reference belongs to a volume/number and standard staff that can be quickly found on the internet (e.g. google scholar).
Please check all parts of the text for consistency and to remove mistakes.
Comments on the Quality of English LanguagePlease the overall check;
Author Response
Please kindly see the attached file. Thanks for your cooperation and understanding.
